# The Significance of Estimated Glomerular Filtration Rate for Predicting Mortality in Glyphosate Herbicide-Intoxicated Patients: A Single-Center, Retrospective Observational Study

**DOI:** 10.3390/jcm11164824

**Published:** 2022-08-17

**Authors:** Seong-Jun Ahn, Jun-Ho Lee, Yong-Hwan Kim, Dong-Woo Lee, Seong-Youn Hwang, Jong-Yoon Park

**Affiliations:** Department of Emergency Medicine, Samsung Changwon Hospital, Sungkyunkwan University School of Medicine, Changwon 630-723, Korea

**Keywords:** glyphosate, herbicide, poisoning, eGFR, mortality

## Abstract

Background: Glyphosate herbicide (GH) is widely used worldwide. It has a higher fatality rate than expected. GH-poisoned cases are increasingly reported. Acute kidney injury in poisoned patients is one of several predictors of GH mortality. The aim of this study was to determine whether estimated glomerular filtration rate (eGFR) could predict kidney injury in GH intoxication. Methods: This was a retrospective study conducted at the emergency department (ED) of a single hospital between January 2004 and December 2021. A total of 434 patients presented with GH intoxication via oral ingestion, and 424 were enrolled. Their demographic characteristics, laboratory variables, complications, and mortality were analyzed to determine clinical predictors associated with GH-induced mortality using a logistic regression analysis. The relationship between GH intoxication and eGFR was determined based on the results of dominance analysis. Additionally, the comparison of creatinine and eGFR was performed through receiver operating characteristic (ROC) curves. Results: A total of 424 GH-poisoned patients were enrolled. Of them, 43 (10.1%) died. In the multivariable analysis, initial GCS (OR: 0.874; 95% CI: 0.765–0.998, *p* = 0.047), albumin (OR: 0.874; 95% CI: 0.765–0.998, *p* = 0.027), pH (OR: 0.002; 95% CI: 0.000–0.037, *p* < 0.001), QTc interval (OR: 1.018; 95% CI: 1.007–1.029, *p* = 0.001), and eGFR (OR: 0.969; 95% CI: 0.95–0.989, *p* = 0.003) were independent factors for predicting in-hospital mortality. In the dominance analysis of the relative importance of the predictive factors, pH accounted for the largest proportion at 35.8%, followed by QTc (20.0%), GCS (17.3%), eGFR (17.0%), and albumin (9.9%). Additionally, eGFR had a larger area under the ROC curve (0.846; 95% CI, 0.809–0.879) than that of creatinine (0.811; 95% CI, 0.771–0.848, *p* = 0.033). Conclusion: In sum, eGFR, considered a surrogate of renal function, was a useful prognostic factor for mortality in glyphosate herbicide-poisoned patients.

## 1. Introduction

Glyphosate is a globally used herbicide that targets the synthesis of chlorophyll-related molecules by the competitive inhibition of the enzyme enolpyruvylshikimate phosphate synthase, following a plant-specific protein synthesis known as the shikimic acid pathway. Because its pathway presents in only plants but not in mammals, glyphosate is expected to have less toxic effects on humans. Reported fatality rates of glyphosate herbicide (GH) poisoning ranged from 3.2% to 29.3% [1]. Since GH is increasingly used among herbicides worldwide, the relatively high toxicity of GH is a problem for the emergency medical system.

Several predictors of mortality associated with GH intoxication have been reported, including old age, large dosage of ingestion, altered mental status, hypotension, respiratory failure, lactic acidosis, and elevated creatinine (Cr) concentration [2,3,4,5,6,7]. Among these variables, serum Cr elevation, if it results from acute kidney injury, might be caused by the direct nephrotoxicity of GH itself, or its indirect effects lead to renal epithelial and tubular cell dysfunctions due to the ischemic insult of hypotension and oxidative stress [8,9,10].

Estimated glomerular filtration rate (eGFR), along with Cr, is being used widely as one of the indicators to evaluate renal function. Unlike absolute serum Cr level, which is affected by individual muscular mass, eGFR is calculated by correcting Cr for gender and age. Thus, it could be more useful for evaluating kidney injury in toxicological cases. Because age and serum Cr have been previously reported as the prognostic predictors of patients with GH poisoning, it is anticipated that the use of eGFR in these patients might be additionally helpful for predicting their severity.

Thus, the objective of this study was to verify the utility of eGFR for predicting the prognosis of GH-poisoned patients visiting the emergency department.

## 2. Methods

A retrospective cohort study was conducted after it was approved by the Institutional Review Board (IRB) of Samsung Changwon Hospital, Sungkyunkwan University School of Medicine (IRB No. SCMC 2022-04-015).

### 2.1. Study Design and Enrollment

GH-intoxicated patients aged ≥ 18 years who visited the emergency department (ED) of our hospital between January 2004 and December 2021 were enrolled. The ED is a regional tertiary emergency medical center with an annual average volume of 45,000 emergency visits. GH intoxication was diagnosed based on the history provided by patients, their family, or emergency medical personnel and from the container label showing the toxic product information. However, no serological screening test of GH was performed for toxicological diagnosis. In this study, the only route of GH poisoning was oral ingestion. Exclusion criteria for enrollment were as follows: (1) unavailable clinical information, (2) co-ingestion with other pesticides or toxic materials except alcohol ingestion, or (3) discharge against medical advice or transfer to other facilities.

### 2.2. Patient Characteristics and Clinical Courses

Patients’ demographic data, history, vital signs, laboratory tests, and their clinical courses were obtained from electronic medical records. A standardized profile of patients was used for the toxicology registry form of the ED. Ingested volumes of GH were recorded as the amount claimed by the patient or guardian or the remaining amount in the bottles and classified approximately based on the suggestion of a previous study as follows [4]: “a spoon” (5 mL), “a mouthful” (25 mL), “a cup” (100 mL), or “a bottle” (300 mL).

In this study, the following cases were defined as complications of the relevant organ system. Pneumonia was diagnosed by the presence of acute pulmonary lesions on chest radiography or chest computed tomography (CT) corresponding to patients’ respiratory symptoms or fever. Acute respiratory failure was defined as a case requiring mechanical ventilation if the patient did not respond to supplement oxygen therapy, if excessive carbon dioxide was not adequately cleared, or if the patient had excessive difficulty with respiration with tachypnea. Rhabdomyolysis was defined as a creatine kinase elevation of >1000 IU/L. Acute kidney injury (AKI) was defined as an elevation of serum creatinine >0.3 mg/dL or 1.5 times the previously recorded baseline creatine level [11]. If the corrected QT interval (QTc) measured using Bazett’s formula (QT interval/root RR) was prolonged above 500 or more milliseconds (ms), significant QT prolongation was considered [12].

### 2.3. Calculations and Classification of Estimated Glomerular Filtration Rate (eGFR)

The widespread adoption of estimated glomerular filtration rate (eGFR) reporting following the publication of the MDRD (Modification of Diet in Renal Disease) equation resulted in the development of a staging system designed to define the severity of the renal impairment and to highlight management concerns at each stage [13,14,15]. It was the first modern-era GFR estimating equation, derived from data on adult patients with predominantly nondiabetic CKD who had their GFR measured at baseline using the urinary clearance of iothalamate [16]. However, the MDRD study equation is mainly derived from White individuals of an average age of 51 ± 12.7 years with nondiabetic kidney disease, and the average GFR 40 mL/min per 1.73 m^2^. Therefore, the MDRD equation is inaccurate, especially in other populations such as patients with nondiabetic kidney disease, diabetes, or liver disease.

Another equation, the Cockcroft–Gault equation, estimates creatinine clearance from the serum creatinine in patients with stable serum creatinine [17]. This formula assumes that creatinine production decreases with aging and is greater in individuals with greater weight. However, this equation was developed when obesity was far less common. In the current era, higher weight may mean greater fat mass and not greater muscle mass.

The 2009 Chronic Kidney Disease Epidemiology Collaboration (CKD-EPI) was developed to provide an accurate estimate of GFR among individuals with normal or only mildly reduced GFR, above 60 mL/min per 1.73 m^2^ [18]. This study population included people with and without kidney disease who had a wide range of GFRs. The 2009 CKD-EPI equation was as accurate as the MDRD study equation among individuals with eGFR less than 60 mL/min per 1.73 m^2^ and somewhat more accurate in those with higher GFRs, but these older equations include a term for race that for any given creatinine value results in a higher eGFR for Black individuals as compared with other individuals.

As a result of these concerns, the CKD-EPI group developed the 2021 CKD-EPI equation for estimating GFR from serum creatinine without a term for race [19]. Compared with the 2009 CKD-EPI creatinine equation, the 2021 equation is slightly less accurate, but it is acceptable for clinical use in many circumstances. Therefore, we calculated eGFR of GH-intoxicated patients according to this guideline published in 2021 and recommended by the National Kidney Foundation. The accurate calculation formula is shown below:eGFR (mL/min/1.73 m^2^) = 142 × (minimum value of serum Cr/K or 1.0)^α^ × (maximum value of serum Cr/K or 1.0)^−0.544^ × 0.9938^Age^ × (1.0 if male or 1.012 if female)

The K constant is 0.9 for males and 0.7 for females. The α constant is −0.302 for males and −0.241 for females. The unit is mg/dL for serum Cr and years for age.

Using the eGFR, we classified the degree of renal impairment into six stages according to the Kidney Disease: Improving Global Outcomes (KDIGO) CKD workgroup guidelines published in 2013 [13]. Each stage of renal injury was classified from grade 1 (G1) to grade 5 (G5) according to the following criteria: G1, normal (eGFR ≥ 90 mL/min/1.73 m^2^); G2, mild reduction (eGFR = 60–89 mL/min/1.73 m^2^); G3a, mild to moderate reduction (eGFR = 45–59 mL/min/1.73 m^2^); G3b, moderate to severe reduction (eGFR = 30–44 mL/min/1.73 m^2^); G4, severe reduction (eGFR = 15–29 mL/min/1.73 m^2^); and G5, renal failure (eGFR < 15 mL/min/1.73 m^2^).

### 2.4. Statistical Analyses

Nominal variables are expressed as frequency with percentages. Continuous variables are expressed as median with interquartile range or mean with standard deviation, depending on their normality of the Kolmogorov–Smirnov test. Pearson’s chi-squared test or Fisher’s exact test for categorical variables and Student’s *t*-test or the Mann–Whitney U test were used to compare groups depending on the normality test. Logistic regression analysis was performed to determine factors related to in-hospital mortality if there was a statistically significant variable after the abovementioned analyses. If collinearity between variables was not suspected, all significant variables from the simple analysis were included in a multiple logistic regression model with backward elimination (Wald) method. Arterial blood gas analysis (ABGA) findings such as pH, bicarbonate, base excess, and lactate seemed to have collinearity, so only pH was included in the logistic regression model considering its statistical and clinical significance.

In addition, we performed dominance analysis to compare the relative importance of the variables included in the final logistic regression model using McFadden’s R^2^ [20]. From its all-subset models, conditional dominance statistics of each variable were calculated and then summed to obtain the general dominance statistics. Relative importance rate was calculated by dividing the general dominance statistic of each variable by the total amount. Finally, we compared discriminant power between serum Cr and eGFR for predicting mortality using receiver operating characteristic (ROC) curve analysis. A *p*-value of less than 0.05 was considered statistically significant. The most statistical analyses were performed with SPSS Statistics ver. 24.0 (IBM Corp., Armonk, NY, USA), but dominance analysis and the comparison of the areas under ROC curves (AUCs) were performed using the ‘dominance analysis’ package in R-project ver. 4.0.1 (R Foundation for Statistical Computing, Vienna, Austria) (https://www.r-project.org, accessed on 20 April 2022) and MedCalc ver. 20.1 (MedCalc Software Ltd., Ostend, Belgium).

## 3. Results

During the study period, 434 patients with GH intoxication visited the ED. Of these, 2 patients were excluded due to their young age (<18 years), and 8 patients were excluded due to missing values, so a total of 424 patients were enrolled in this study (Figure 1).

### 3.1. Patients’ Characteristics and Comparison of Laboratory Measurements

The patients’ characteristics such as demographics, vital signs, and laboratory data are presented in Table 1, in which after dividing into two groups, survivor (SG) vs non-survivor (NSG), their characteristics were compared with each other. The NSG was significantly older than the SG, and the amount of ingestion was larger in the NSG (*p* < 0.001), but the time interval from GH ingestion to ED arrival did not differ between the two groups (*p* = 0.875). Initial systolic blood pressures (SBP) in the NSG were lower than those in the SG (*p* < 0.001) and heart rates were not significantly different (*p* = 0.072). The Glasgow Coma Scale (GCS) score for the NSG was 9 ± 4, which was significantly (*p* < 0.001) lower than that (14 ± 3) for the SG. Gastrointestinal decontamination with gastric lavage and administration of charcoal were not significantly different between the two groups.

Laboratory blood tests were conducted for statistically significant differences in the following parameters between the two groups: white blood cell count (WBC, *p* < 0.001), creatinine (Cr, *p* < 0.001), albumin (*p* < 0.001), aspartate aminotransferase (AST, *p* = 0.004), glucose (*p* < 0.001), sodium (Na^+^, *p* < 0.001), potassium (K^+^, *p* = 0.004), amylase (*p* = 0.004), lipase (*p* = 0.008), pH (*p* < 0.001), bicarbonate (HCO_3_^−^, *p* < 0.001), base excess (*p* < 0.001), and lactate (*p* < 0.001). Among these variables, levels of K^+^, pH, HCO_3_, base excess, and lactate were particularly noteworthy. The NSG showed a tendency of higher K^+^ and lactate levels but lower pH, HCO_3_, and base excess than the SG. In the case of eGFR classification, grade 5 was the most common in the SG while grade 4 was the most common in the NSG (*p* < 0.001). After analyzing initial ECG in the ED, QTc intervals were calculated. Those in the NSG were more prolonged than those in the SG (NSG: 526 ± 50.5 msec vs. SG: 464.9 ± 36.3 msec, *p* < 0.001). All detailed laboratory variables and their *p*-values are provided in Table 1.

### 3.2. Complications of Glyphosate-Poisoned Patients Group According to in-Hospital Mortality

In this study, 381 patients among 434 poison cases survived. The overall in-hospital mortality rate was 10.14%. The most frequent complication was metabolic acidosis (50.0%), followed by hypotension (45.3%), respiratory failure (45.3%), pneumonia (42.5%), acute pancreatitis (41.5%), acute kidney injury (40.1%), hyperkalemia (38.2%), rhabdomyolysis (38.2%), and seizure (35.4%). The median duration of ventilator care was significantly longer in the NSG (NSG: 4.1 ± 8.8 days vs. SG: 1.1 ± 3.9 days, *p* = 0.039). Patients in the SG stayed longer in the ICU than those in the NSG, although their difference was not statistically significant (*p* = 0.210). Hemodialysis was conducted for eight patients in the SG and six patients in the NSG (*p* = 0.001). These results are summarized in Table 2.

### 3.3. Multivariable Analysis of Factors Associated with in-Hospital Mortality

For constructing the multiple logistic regression model, significant variables in the simple analysis were selected from the demographic patterns, vital signs, and laboratory findings of the patients. However, only pH among the ABGA findings was included in the statistical regression model considering their mutual collinearity. As a result, the variables included in the final model were as follows: age, amount of ingestion, SBP, HR, GCS, WBC, AST, albumin, glucose, Cr, pH, QTc interval, and eGFR, and GCS, albumin, pH, QTc interval, and eGFR were significantly related factors for predicting in-hospital mortality (Table 3). Among them, pH showed the most correlation with mortality, but QTc interval and eGFR also showed significant associations with mortality. In the case of Cr, its significance disappeared in the multiple regression analysis. However, in the case of GFR, its statistical significance was maintained in the multivariate regression analysis.

### 3.4. Dominance Analysis Amongst Predictors of In-Hospital Mortality

Dominance analysis was performed on the five variables included in the final statistical model using McFadden’s R^2^ statistic. Their additional R^2^ contributions, that is, conditional dominances, in all subset models are shown in Figure 2. The general dominance summing the conditional dominance statistics and the relative importance are presented in Figure 3. As shown in Figure 3A, pH had the largest general dominance statistic of 0.209, which accounted for 36% of the total R^2^ of 0.584. The QTc interval, GCS and eGFR showed similar general dominance statistics between 0.10 and 0.12, which corresponds to relative importance between 17% and 20%. Albumin has the smallest relative importance of only 10%. Therefore, initial pH was the most influential predictor of mortality. Among the remaining variables, QTc interval, GCS, and eGFR had the next most influence on mortality prediction. Albumin was the fifth-ranked mortality predictor, with the least influence.

Meanwhile, a model including creatinine instead of eGFR was also analyzed. McFadden’s R^2^ for this model was 0.578, a slight decrease compared with the above model. In this model, creatinine had the smallest dominance statistics, irrespective of whether it was conditional or general, and the relative importance rate was only 5%, which was a drop of more than 10% compared with eGFR (Figure 2B and Figure 3B).

### 3.5. ROC Curve Analysis of eGFR and Serum Creatinine

ROC curve analysis was performed to compare the discriminant power between eGFR and serum Cr for predicting mortality (Figure 4). The area under the curve (AUC) of 0.846 (95% CI: 0.809–0.879) for the eGFR was larger than that of 0.811 (95% CI: 0.771–0.848) for serum Cr (*p* = 0.033), and the eGFR compared with serum Cr showed better discrimination for predicting mortality.

## 4. Discussion

In this study, the eGFR of GH-poisoned patients was an independent predictor for in-hospital mortality in addition to other predictors such as initial GCS, albumin, pH, and QTc interval. Although the relative importance of eGFR was lower than that of pH, it had almost the same importance as both QTc and GCS. Moreover, Cr did not show statistical significance unlike the eGFR. That is, eGFR showed more discriminatory power than serum Cr for estimating the prognosis of GH-poisoned cases.

Although GH works by inhibiting the pathway of plant-specific enzymes, GH-poisoned patients develop various complications that could lead to death [21]. Many previous studies have reported complications of GH such as cardiovascular instability, respiratory distress, metabolic acidosis, and renal dysfunction [5,6,22,23,24]. Similar complications were also found in our study. Additionally, several studies have investigated factors for predicting death due to GH toxicity. Old age, large dosage of ingestion, altered mental status, shock, respiratory failure, metabolic acidosis, elevated Cr concentration, and QTc prolongation have been reported as predictors of mortality [2,6,25,26]. These reported predictors were similar to the predictors found in the present study.

Serum Cr has been mentioned as one of the prognostic factors in previous studies. However, Cr showed no statistically significant association with in-hospital mortality in the present study. On the other hand, eGFR was highly associated with mortality, similar to GCS, a predictor that researchers would expect logically.

Acute kidney injury could develop in GH-poisoned cases. The toxicological mechanism may be hypovolemic shock due to secondary systemic toxicity caused by GH. In addition, direct renal toxicity of GH has been reported based on histologic findings of renal epithelial injuries in proximal tubules [27,28]. In vitro studies have revealed pathologic changes of cells from Bowman’s space to convoluted tubules and lymphocytic infiltration necrosis with deformation of nephrons in GH-intoxicated animal cells [29,30]. It has been reported that AKI caused by GH was different from that typically seen in ischemic acute tubular injury [31]. In AKI caused by GH, large vacuoles were observed in the proximal tubular epithelium, suggesting mitochondrial toxicity due to glyphosate. In addition, glomerular subendothelial edema and podocyte alteration were also observed due to vascular endothelial injury. The cause of this phenomenon is presumed to be the combination effect of glyphosate and its surfactant rather than renal ischemia caused by hypotension.

Creatinine, a breakdown product of creatine phosphate, is excreted via the kidney. Serum creatinine level has been reported to reflect the glomerular filtration rate [32]. Although Cr is the most widely used marker for evaluating renal function including AKI, it has several limitations. First, serum Cr does not immediately reflect the onset of renal damage because it not only changes depending on physiological factors such as age and patient’s muscle mass but also has a time delay until its increase after a kidney injury has occurred. In patients with previously normal renal function, serum Cr elevation might not appear until more than half of the GFR is reduced due to large reserves in the kidney [33]. In case of GH-induced nephrotoxicity, serum Cr is also the most common biomarker for evaluating renal function in previous studies. For the above-mentioned reasons, several studies have reported the usefulness of other laboratory tests other than Cr for predicting AKI in GH-intoxication. Such biomarkers include urinary kidney injury molecule-1 (uKIM-1) and urinary cystatin c (uCysC). However, it is difficult to use them widely clinically in toxicological cases because they are experimental laboratory tests [10,34]. On the contrary, eGFR is an easily available item that can be calculated mainly from serum Cr. In addition, there is no burden of additional blood testing as it is a corrected value calculated by adapting gender and age. Thus, eGFR might be more useful for evaluating AKI than serum Cr. In fact, other studies on diagnosing and predicting AKI have reported that eGFR is superior to serum Cr [35].

On the other hand, it is interesting that albumin was found to be a significant predictor of mortality in this study. Albumin is produced by the liver, but it is known that it is also affected by several other causes such as chronic illness, heart failure, and malnutrition [36]. As shown in Table 1, liver disease was not associated with mortality, so deterioration of underlying health status from the other causes that can lower albumin may have contributed to the death.

This study has several limitations. First, this was a retrospective study. Each treatment was performed individually for GH-poisoned patients, which might have led to differences in mortality rates. However, there is no specific treatment for GH intoxication in our ED. Thus, bias due to this would be limited. Second, serum GH concentration could not be measured in toxic cases. In addition, the actual ingestion dose of GH was not directly measured. There might be mismatched ingestion dosage. Third, due to the absence of the measurement of serum GH concentration as mentioned above, the diagnosis of GH poisoning might be wrong because it relied on patients’ or guardians’ statements when they brought the pesticide container. Fourth, although there are KIM-1 and NGAL, which are more sensitive biomarkers for detecting AKI, we could not use these new biomarkers because those biomarkers were not available at our hospital in the period of study. Fifth, since the GH poisoning patients in this study were those who visited a single hospital, it is thought that it is difficult to generalize their outcomes as representative of all GH-poisoned patients. Sixth, although it is reported that alcohol may affect the measurement of serum creatinine [37], this study only confirmed whether the GH-poisoned patients consumed alcohol together with the GH, but we did not measure the serum ethanol concentration itself. There is another disadvantage that the serum Cr affected by co-ingested alcohol could not be identified. Lastly, in this study, we analyzed only initial blood tests for GH-poisoned patients. Thus, there was a disadvantage in that we could not estimate their prognosis using consecutively performed laboratory tests during follow-up.

## 5. Conclusions

In this study, eGFR had an independent relationship with in-hospital mortality as a predictor, in contrast with serum Cr. Further studies are needed to prove the usefulness of eGFR for managing GH poisoned patients in ED.

## Figures and Tables

**Figure 1 jcm-11-04824-f001:**
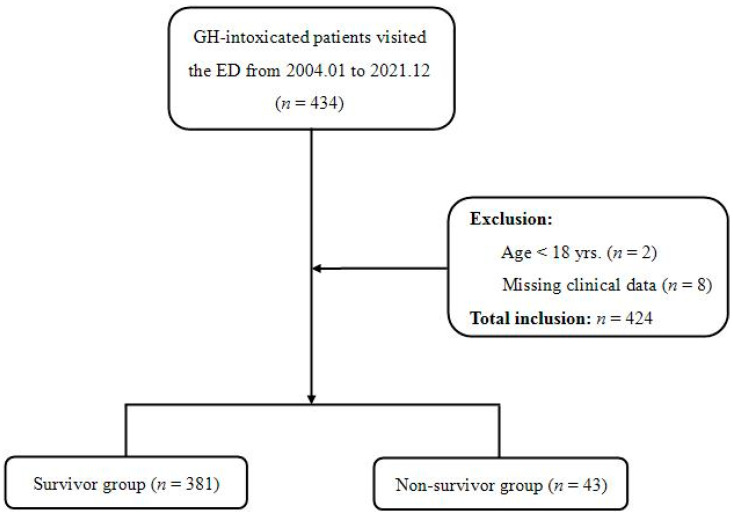
Study flow diagram.

**Figure 2 jcm-11-04824-f002:**
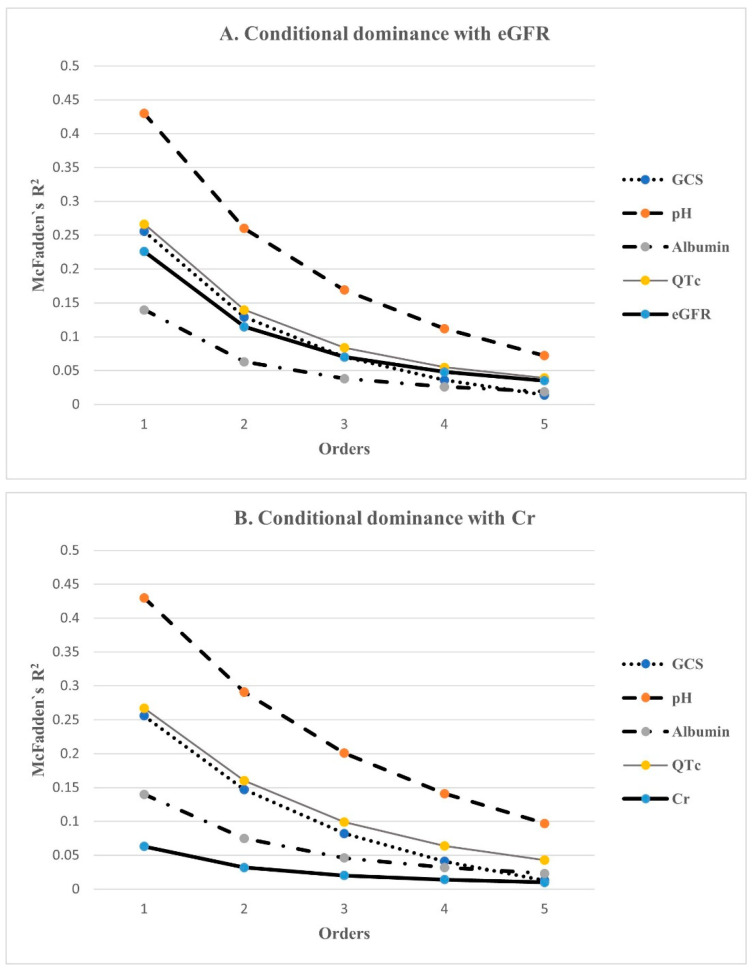
The conditional dominance of independent factors for predicting in-hospital mortality. Conditional dominance statistics of each variable were derived from averaging the 5 within-order subsets (1–5) of their R^2^s. The values tended to decrease as the level elevated. Among them, pH had the highest value in both analyses, while albumin had the lowest value in (**A**), and Cr was the lowest in (**B**). QTc, eGFR, and GCS had almost the same values. GCS: Glasgow Coma Scale; QTc: corrected QT interval; eGFR: estimated glomerular filtration rate; Cr: creatinine.

**Figure 3 jcm-11-04824-f003:**
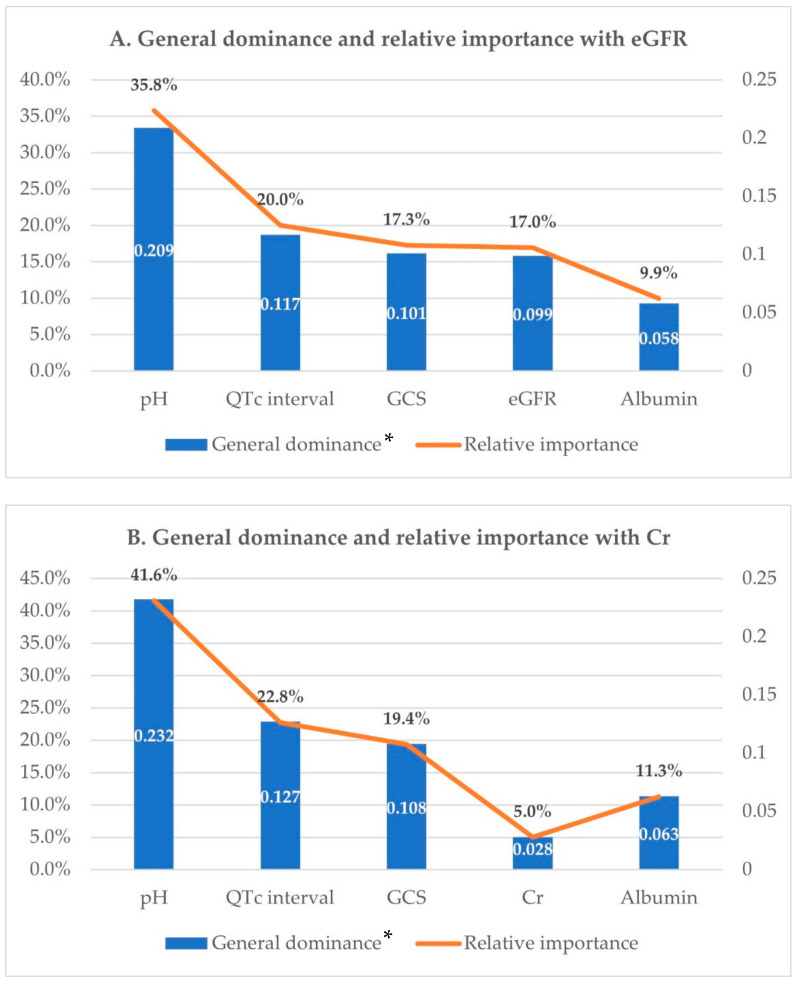
Dominance analysis. General dominance and relative importance of independent factors for predicting in-hospital mortality. GCS: Glasgow Coma Scale; QTc interval: corrected QT interval; eGFR: estimated glomerular filtration rate; Cr: creatinine. * Calculated by summing the conditional dominance statistics for each predictor.

**Figure 4 jcm-11-04824-f004:**
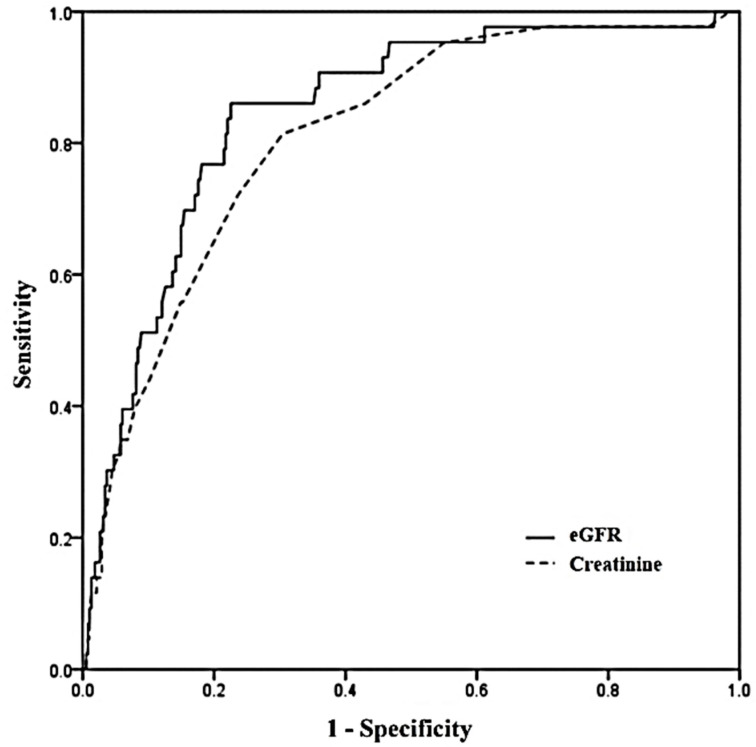
Receiver operating characteristic (ROC) curves for eGFR and creatinine. Areas under the curve with 95% confidence intervals was 0.846 (0.809–0.879) for eGFR and 0.811 (0.771–0.848) for creatinine (*p* = 0.033). eGFR: estimated glomerular filtration rate.

**Table 1 jcm-11-04824-t001:** Characteristics and laboratory findings of glyphosate-poisoned patients with in-hospital mortality.

Characteristics	Survivors (*n* = 381)	Non-Survivors (*n* = 43)	*p*-Value
Age (years)	58 (43–73)	70 (59–81)	<0.001
Male	276 (72.4%)	32 (74.4%)	0.783
Estimated ingestion time (min)	192 (0–459)	198 (0–408)	0.875
Amount (mL)	157 (34–279)	241 (112–369)	<0.001
Past Medical History			
HTN	43 (11.3%)	6 (14.0%)	0.879
DM	31 (8%)	5 (11.6%)	0.530
Liver disease	4 (1.0%)	1 (2.3%)	0.323
Vital signs			
SBP (mmHg)	120 (93–148)	82 (41–123)	<0.001
HR (beats/min)	88 (69–107)	78 (42–114)	0.072
RR (breaths/min)	19 (16–22)	18 (8–28)	0.390
BT (°C)	36.5 (35.9–37.2)	36.0 (35.2–36.8)	<0.001
MAP (mmHg)	90 (69–111)	60 (28–93)	<0.001
SaO_2_ (%)	96 (91–101)	85 (64–105)	0.001
GCS	14 (11–16)	9 (4–13)	<0.001
Intentional poisoning	346 (90.8%)	40 (93.0%)	0.631
Co-ingestion with alcohol	105 (27.6%)	6 (14.0%)	0.054
Gastric lavage	321 (84.3%)	34 (79.0%)	0.383
Activated charcoal	191 (50.1%)	23 (53.5%)	0.676
Laboratory findings			
WBC (cells/mL)	11,613 (5860–17,367)	18,075 (10,399–25,751)	<0.001
Hct (%)	40.9 (35.0–46.8)	40.7 (34.2–47.2)	0.846
Platelet (cells/μL)	250.6 (180.2–321.0)	255.0 (158.4–351.6)	0.772
BUN (mg/dL)	15.1 (7.4–22.8)	17.3 (7.5–27.0)	0.168
Cr (mg/dL)	1.0 (0.3–1.7)	1.7 (0.7–2.6)	<0.001
Albumin (g/dL)	4 (3.6–4.8)	3.5 (2.9–4.2)	<0.001
AST (U/L)	39 (4–73)	64 (11–117)	0.004
ALT (U/L)	28 (7–48)	37 (3–72)	0.076
Total bilirubin (mg/dL)	0.8 (0.3–1.3)	0.7 (0.3–1.0)	0.128
Glucose (mg/dL)	152 (84–219)	217 (132–303)	<0.001
Na (mmol/L)	140 (136–145)	144 (138–151)	<0.001
K (mmol/L)	4.1 (3.2–5.0)	4.7 (3.3–6.2)	0.004
Amylase (U/L)	133 (0–291)	251 (4–499)	0.004
Lipase (U/L)	52 (0–108)	118 (0–265)	0.008
CRP (mg/dL)	8.4 (0–27)	15.3 (0–50.4)	0.328
pH	7.35 (7.24–7.47)	7.04 (6.85–7.23)	<0.001
PO_2_ (mmHg)	88.9 (20.7–149)	84.9 (20.7–149)	0.685
PCO_2_ (mmHg)	34 (25–43)	38 (21–55)	0.128
HCO_3_^−^ (mmol/L)	19.5 (15.4–24.5)	12.8 (9.0–16.7)	<0.001
Lactate (mmol/L)	3.1 (0.9–5.3)	6.3 (2.9–9.7)	<0.001
QTc interval (ms)	464.9 (428.6–501.1)	525.5 (475.1–576.0)	<0.001
HD	8 (2.1%)	6 (14.0%)	0.001
ICU admission	198 (52.0%)	24 (55.8%)	<0.001
eGFR (mL/min/1.73 m^2^)	91.4 (70.9–105.3)	50.5 (32.6–63.9)	<0.001
eGFR classification	381 (89.9%)	43 (10.1%)	<0.001
G5	3 (0.8%)	1 (2.3%)	
G4	8 (2.1%)	8 (18.6%)	
G3b	11 (2.9%)	6 (14.0%)	
G3a	35 (9.2%)	12 (27.9%)	
G2	128 (33.6%)	14 (32.6%)	
G1	196 (51.4%)	2 (4.7%)	

Each value is expressed as median (interquartile range). HTN: hypertension; DM: diabetes mellitus; SBP: systolic blood pressure; HR: heart rate; RR: respiratory rate; BT: body temperature; MAP: mean arterial pressure; GCS: Glasgow Coma Scale; WBC: white blood cell count; Hct: hematocrit; BUN: blood urea nitrogen; Cr: creatinine; AST: aspartate aminotransferase; ALT: alanine aminotransferase; CRP: C-reactive protein; QTc: corrected QT interval; HD: Hemodialysis; ICU: Intensive Care Unit; eGFR: estimated glomerular filtration rate. *p*-value < 0.05 is statistically significant.

**Table 2 jcm-11-04824-t002:** Complications of glyphosate-poisoned patients with in-hospital mortality.

Characteristics	Survivors (*n* = 381)	Non-Survivors (*n* = 43)	*p*-Value
Hypotension	192 (50.4%)	41 (95.3%)	<0.001
Respiratory failure	192 (50.4%)	39 (90.7%)	<0.001
Pneumonia	180 (47.2%)	29 (67.4%)	0.012
Acute kidney injury	170 (44.6%)	36 (83.7%)	<0.001
Hemodialysis	8 (2.1%)	6 (14.0%)	0.001
Hyperkalemia	162 (42.5%)	29 (67.4%)	0.002
Metabolic acidosis	212 (55.6%)	42 (97.7%)	<0.001
Rhabdomyolysis	162 (42.5%)	25 (58.1%)	0.051
QTc prolongation	51 (13.4%)	30 (69.8%)	<0.001
Seizure	150 (39.4%)	24 (55.8%)	0.038
Acute pancreatitis	176 (46.2%)	33 (76.7%)	<0.001
Ventilator care days	1.1 (0–5.0))	4.1 (0–12.9)	0.039
Length of ICU stay (days)	3.1 (0–8.1)	2 (0–13.2)	0.210

Each value is expressed as median (interquartile range). QTc: corrected QT interval; ICU: intensive care unit. *p*-value < 0.05 is statistically significant.

**Table 3 jcm-11-04824-t003:** Multivariable analysis of factors associated with in-hospital mortality.

Variables	Unadjusted OR (95% CI)	*p*-Value	Adjusted OR (95% CI)	*p*-Value
Age	1.062 (1.035–1.089)	<0.001	1.023 (0.977–1.071)	0.342
Amount of ingestion	1.005 (1.002–1.007)	<0.001	1.002 (0.998–1.006)	0.332
SBP	0.961 (0.949–0.973)	<0.001	1.003 (0.984–1.022)	0.773
HR	0.978 (0.964–0.993)	0.003	0.989 (0.969–1.009)	0.266
GCS	0.705 (0.646–0.769)	<0.001	0.874 (0.765–0.998)	0.047
WBC	1.000 (1.000–1.000)	<0.001	1.000 (1.000–1.000)	0.746
AST	1.011 (1.005–1.017)	<0.001	0.998 (0.986–1.011)	0.790
Albumin	0.208 (0.122–0.354)	<0.001	0.451 (0.223–0.913)	0.027
Glucose	1.009 (1.005–1.012)	<0.001	0.996 (0.990–1.002)	0.234
Cr	2.081 (1.403–3.087)	<0.001	0.859 (0.461–1.602)	0.634
pH	0.000 (0.000–0.000)	<0.001	0.002 (0.000–0.037)	<0.001
Lactate	1.474 (1.310–1.658)	<0.001	1.145 (0.952–1.377)	0.152
QTc interval	1.035 (1.025–1.045)	<0.001	1.018 (1.007–1.029)	0.001
eGFR	0.952 (0.939–0.965)	<0.001	0.969 (0.950–0.989)	0.003

SBP: systolic blood pressure; HR: heart rate; GCS: Glasgow Coma Scale; WBC: white blood cell count; AST: aspartate aminotransferase; Cr: creatinine; QTc: corrected QT interval; eGFR: estimated glomerular filtration rate. *p*-value < 0.05 is statistically significant.

## Data Availability

The datasets generated and/or analyzed during the current study are not publicly available but are available from the corresponding author on reasonable request.

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
