# Peer review of "The Significance of Estimated Glomerular Filtration Rate for Predicting Mortality in Glyphosate Herbicide-Intoxicated Patients: A Single-Center, Retrospective Observational Study"

_jcm, 2022, doi:10.3390/jcm11164824_

Round 1

Reviewer 1 Report

dear authors,

this is an interesting study , required minor revision ,

1- in the abstract, the patients were recruited not enrolled 

2- in the introduction, explain briefly the mechanism of glyphosate herbicide

3- in the method, estimation of eGFR by this equation is not appropriate , please compare it with other equations

4- in the results, consort-flow diagram is needed

all figures and tables are good

5- in the discussion , please explain GH-induced AKI regarding molecular levels

please explain why you not used other biomarkers like KIM-1 and NGAL which are more specific in detecting AKI than Cr .

i suggest the following references to be cited to increase impact of the paper

Al-Kuraishy HM, Al-Gareeb AI. Acute kidney injury and COVID-19. The Egyptian Journal of Internal Medicine. 2021 Dec;33(1):1-5.

Rasheed HA, Al-Naimi MS, Hussien NR, Al-Harchan NA, Al-Kuraishy HM, Al-Gareeb AI. New insight into the effect of lycopene on the oxidative stress in acute kidney injury. International Journal of Critical Illness and Injury Science. 2020 Sep;10(Suppl 1):11.

Reviewer 2 Report

Dear Editor and authors for journal of clinical medicine

This is an interesting study compare serum Cr and eGFR as surrogate marker to predict mortality for 424 patients with glyphosate herbicide intoxication from 2004 to 2021 in Korea.

By comparing survivors with non-survivors, they concluded several significant factors which decide the fate of patients. However, several points must be considered.

Major issue:

First of all, eGFR is a function of serum Cr and patient’s age. Both are significant factors affect the outcome. So it is no surprising that eGFR, the combination of them, is also a significant factor. But eGFR requires a much more complicated equation to calculate the result.

Furthermore, in table 3, serum Cr and eGFR are both included in the multivariate logistic regression. I will suggest the author repeat another regression with only serum Cr included. Serum Cr will be a significant factor in my expectation because Cr and eGFR are highly related.

In authors’ words, mutual collinearity exists in pH and bicarbonate. This is the same situation for serum Cr and GFR. The ROC curves in figure 2 also reflect the similar result. I did not see a “significant” difference between serum Cr and eGFR. (If there is any, please state it specifically.)

 The reason why eGFR may be a better indicator than serum Cr alone could be eGFR contains a small portion of age and sex factor. And (older) age is highly related to (lower) albumin in clinical observation. So the novelty of this study is not very high.

Maybe a more specific question for this study should be” Can eGFR alone predict better than the combination of serum Cr and age (Albumin)”?

 How to pick the most powerful factor among factors with mutual collinearity (ex: pH and lactate, Age and albumin, Cr and eGFR) follows certain rules. Please consult this with statistical experts.

The authors may repeat the analysis of Figure 1A and 1B with serum Cr instead of eGFR to see what result they may found eGFR. This may persuade readers the importance of eGFR if serum Cr can’t replace eGFR.

Please also re-arrange the order of the meaning by each different line with highest to lowest conditional dominance at the right part of figure 1-A.

The data of acute kidney injury is available in Table 2. May be the increase of serum Cr may represent a more important factor to predict mortality than initial serum Cr alone.

Such protocol is frequently used to predict mortality in acute pancreatitis as Ranson criteria. Since pancreatitis occurred to almost half of the enrolled patients by table 2, it may be an interesting topic to investigate.

Other minor issues

1.      How many patients are transferred out during the enrollment period under the same diagnosis? This might affect the result since this is a single center study results from selection bias. The location and facilitation of the hospital may influence the ER doctors’ decision to transfer intoxicated patients if there is a tertiary medical center nearby. Authors may mention the related information in the limitation.

2.      Also, for patients who are intoxicated, alcohol, glucose and other organic compounds may affect the accuracy of serum Cr value. This may cause measurement bias in the first place. [1]

In Table 2, non-survivors are older and with much more comorbidity. What about the percentage of diabetes over these two groups?

How many of them are also intoxication with alcohol?

3.      English polishing may be needed to check typos and grammar errors. For example, page 10, line 319, “in adition” should be “in addition”.

4.      The explanation for eGFR classification and hemodialysis should be more detailed.

I.            What is the meaning of p<0.001 for each grade between two groups in table 1?

II.          Also, what is the meaning of hemodialysis in Table 2? Are these patients already under chronic dialysis or they received urgent dialysis due to intoxication?

In summary, it is a well-organized study. I will suggest major revision and submit again.

Reference

1.         Liu, W.S.; Chung, Y.T.; Yang, C.Y.; Lin, C.C.; Tsai, K.H.; Yang, W.C.; Chen, T.W.; Lai, Y.T.; Li, S.Y.; Liu, T.Y. Serum creatinine determined by Jaffe, enzymatic method, and isotope dilution-liquid chromatography-mass spectrometry in patients under hemodialysis. J Clin Lab Anal 2012, 26, 206-214, doi:10.1002/jcla.21495.

Round 2

Reviewer 2 Report

Q: Table 1, we conducted the analysis by performing the chi-squared test. Since this is a simple regression analysis, the p-value is shown for eGFR classification, not for each grade. Does the reviewer mentioned by any chance mean multiple comparisons comparing each grade based on grade 1 or grade 5?

A: Yes, I mean it. Please conduct the comparison on CKD grade 1 to grade 5.
